# Studies on Pollen Morphology, Pollen Vitality and Preservation Methods of *Gleditsia sinensis* Lam. (Fabaceae)

**Qiao Liu** [1,2,3,4], **Ju Yang** [1,2,3,4], **Xiurong Wang** [1,*] **and Yang Zhao** [1,2,3,4]

1   College of Forestry, Guizhou University, Guiyang 550025, China
2   Institute for Forest Resources & Environment of Guizhou, Guizhou University, Guiyang 550025, China
3   Key Laboratory of Forest Cultivation in Plateau Mountain of Guizhou Province, Guiyang 550025, China
4   Key Laboratory of Plant Resource Conservation and Germplasm Innovation in Mountainous Region (Ministry of Education), Guizhou University, Guiyang 550025, China
*   Correspondence: xrwang@gzu.edu.cn

**Abstract:** *Gleditsia sinensis* Lam. (Fabaceae) is an endemic species in China, which has a wide range of ecological functions and high economic value. *G. sinensis* belongs to androdioecy, and the stamens of perfect flowers are aborted, meaning that a perfect flower is a functional female flower. Understanding the dynamic process of flowering and the characteristics of pollen morphology effectively determine the viability of pollen vitality, and the suitable conditions for short-term storage of pollen can provide theoretical basis and technical reference for hybrid breeding and germplasm conservation of *G. sinensis.* In this study, the male plants of *G. sinensis* in Guiyang area were used as research materials. The flowering dynamic process of male flowers was recorded through field observation. The morphology of pollen was observed and analyzed with a scanning electron microscope (SEM). The germination characteristics of pollen were studied with an in vitro germination method, and the pollen vitality was also determined using four staining methods. The effects of different storage temperatures and water contents on pollen germination rate were discussed. The results showed that the male flowers of *G. sinensis* had a short, single flowering period, lasting 2–3 days from the opening to the shedding. The dynamic opening process of a single flower was artificially divided into five stages. Pollen grains of *G. sinensis* are oblate spheroidal, tricolporate with equatorial elongated endoapertures and the sporoderm surface is reticulate. The MTT (Thiazolyl Blue Tetrazolium Bromide) staining method could accurately and quickly determine the pollen vitality of *G. sinensis*. The highest pollen germination rate was 65.89% ± 3.41%, and the length of the pollen tube was 3.96 mm after cultured in 15% sucrose + 100 mg/L boric acid + 20 mg/L calcium chloride for 24 h. It was necessary to collect the pollen at the big bud stage, which was conducive to improving the efficiency of pollen collection because the pollen had been mature with high pollen vitality at this stage. When it came to pollen preservation, the pollen germination rate was significantly affected by storage time, storage temperature and pollen water content. The pollen still had high vitality after being stored at −80 °C for 30 days when the moisture content of the pollen decreased to 9%, and the pollen germination rate only decreased by 28.84% compared with that before storage. In conclusion, this study has comprehensively and systematically studied the morphology, vitality determination and preservation methods of the pollen of *G. sinensis*, providing a theoretical basis for the cross regional breeding and the conservation and utilization of germplasm resources.

**Keywords:** *Gleditsia*; pollen viability; pollen germination; pollen preservation



## 1. Introduction

In seed plants, pollen is of great importance for sexual reproduction as a male gametophyte containing genetic information [1]. Pollen viability is a prerequisite for successful pollination, playing an important role in fertilization and fruit setting of flowering plants [2,3], embryo development [4,5], seed quality [6,7] and breeding efficiency. Due to

this importance, researchers have conducted many studies on pollen, such as the pollen development process [8], electron microscopy palynology [9], pollen viability detection and pollen preservation methods [10,11], signal transduction and related regulation during pollen tube growth [12–14]. All studies above are indispensable for plant cross breeding from practice to theory [15].

*Gleditsia sinensis* Lam. (Fabaceae) belongs to genus *Gleditsia* in Fabaceae, which is an endemic species widely distributed in China [16,17]. It is an ideal tree for economic forest, timber forest, shelterbelt and landscaping because of its tall and beautiful shape, strong resistance and barren resistance. The extract of saponin and pod is an excellent Chinese medicinal material [18] and industrial raw material [19], which has high application value and broad prospect. *G. sinensis* belonged to androdioecy, and its bisexual flowers are functionally female, with most stamens aborted [20] and with a breeding system dominated by outcrossing [21]. In this respect, the fruit and seed set are almost entirely dependent on the pollens provided by male plants during synchronous flowering. Dioecious plants have been under great reproductive pressure due to frequent occurrences of missed flowering periods, lack of pollinators, and imbalanced sex ratios in recent years [22,23]. This necessitates the conservation of pollen resources. In the past, studies of *G. sinensis* focused mostly on its medicinal components [18], cultivation techniques [24], germplasm resource investigation [25] and stress physiology [26,27] while only a few studies examined its reproductive process [28] like the pollen morphology observed [29] and observation of flowering characteristics [30]. In contrast, pollen-related research of *G. sinensis* is fragmented and largely unknown, which is not conducive to the popularization and application of this plant.

Therefore, the observation of pollen morphological characteristics by scanning electron microscope, investigation of single male flower flowering dynamic process, optimized in vitro germination and the effects of different storage conditions on pollen preservation have been analyzed in this study, and the results based on the above studies are supposed to provide reference for the research of the reproductive biology of *G. sinensis*, the preservation of germplasm resources and the hybridization and breeding.

## 2. Materials and Methods

### 2.1. Study Site

Experiments were conducted at Guiyang's planting base (26.45 N, 106.56 E) from March to May of 2021 and 2022. In the experiment field, the average annual temperature was 15.7 °C, and the average annual precipitation was 1215.7 mm. Our test plants were twelve-year-old male plants that had consistently grown and flowered in the plantation.

### 2.2. Observation of Male Single Flower Flowering Dynamic Process

A study of the flowering dynamics of a male single flower of *G. sinensis* was conducted in 2021 and 2022. A total of 30 male flowers at the full bud stage were placed on each test plant, and 6 trees at full bloom stage were employed for observation in total. We observed and photographed the flower opening process several times over the course of several days, from 8:00 a.m. to 18:00 p.m., and the viability of pollen was determined by collecting pollen at corresponding stages.

### 2.3. Pollen Morphology Observation

#### 2.3.1. Light Microscopic Studies

Mature anthers collected were preserved in glass vials containing acetic alcohol (1:3). The material was then centrifuged and washed twice with distilled water to remove traces of acetic alcohol. After this, acetolysls after preservation in glacial acetic acid was used as recommended by Reitsma [31] and Raynal A and Raynal J [32]. The material was given three washings, and residual material was mounted in glycerin jelly that was already stained in 1% safranine as recommended by Kisser [33] and Erdtman [34]. In the Leica research microscope, 5–7 slides were prepared to study. The measurement of pollen characters was

made from 50 grains taken at random. Microphotographic work was carried out using Leica DM300 Photomicroscope fixed in Institute for Forest Resources & Environment of Guizhou, Guizhou University. The terminology used was in accordance with the studies by Erdtman [34] and Halbritter [35].

2.3.2. Scanning Electron Microscopy

The anthers of newly opened male flowers on the test plants were randomly selected at the peak of the flowering period in 2022 and then soaked in glutaraldehyde solution (prepared with pH 8.0 phosphate buffer) at 4 °C for 24 h. The fixed pollen grains were then dehydrated and dried with an acetone series (30%, 50%, 70%, 80%, 90%, 95%, 100% and 100%). Afterward, they were mounted on metallic stubs, coated with gold-palladium, and then observed under a scanning electron microscope (SEM). A Hitachi JSM-SU8100 scanning electron microscope was used to take photos of pollen grains.

*2.4. Study on Methods for Determining Pollen Viability*
2.4.1. In Vitro Germination of *G. sinensis*
The Effects of Different Medium Components on the In Vitro Germination of *G. sinensis*

Using a single factor experimental design (Table 1), we examined the influence of sucrose, boric acid ($H_3BO_3$) and $CaCl_2$ on pollen germination to determine the concentrations of media components that are conducive to in vitro germination. The specific operation was as follows: In a 2 mL centrifuge tube, fresh anthers were mixed with 1 mL of different liquid medium formulations. After homogenizing shock, anthers were filtered for pollen suspensions, which were incubated for 4 h at a constant temperature of 25 °C. A suspension of cultured pollen was mixed and dropped onto the slide. Once covered, the slide was observed under a microscope and photographed. For each slide, five visual fields were randomly selected, and each treatment was repeated 6 times. The length of pollen tubes at germination exceeded the diameter of pollen grains was used to judge pollen germination. The following formula was used to calculate the in vitro germination rate of *G. sinensis*: pollen germination rate (%) = germination pollen number/observed total pollen number × 100.

**Table 1.** In vitro pollen germination media.

| Media Type | Control (CK) | Sucrose (%) | | | | Boric Acid (mg/L) | | | | Calcium Chloride (mg/mL) | | | |
|---|---|---|---|---|---|---|---|---|---|---|---|---|---|
| Sucrose | 0 | 5 | 10 | 15 | 20 | | | | | | | | |
| Boric acid | 0 | | | | | 50 | 100 | 150 | 200 | | | | |
| Calcium chloride | 0 | | | | | | | | | 50 | 100 | 150 | 200 |
| Distilled water | 100 mL | | | | | | | | | | | | |

Screening of Pollen Liquid Medium

Three factors and three levels (Table 2) were selected based on the concentration of media components tested by a single factor test. In addition, an orthogonal test (L9 (34)) was designed to identify the medium formula best suited for determining pollen in vitro germination rates. The specific operation was the same as described in single factor experimental, except extending the culture time to 12 h.

**Table 2.** Factor levels of liquid medium screening for in vitro germination of *G. sinensis*.

| Levels | Sucrose (%) A | Test Factors Boric Acid (mg/L) B | Calcium Chloride (mg/L) C |
|---|---|---|---|
| 1 | 5 | 50 | 20 |
| 2 | 10 | 100 | 40 |
| 3 | 15 | 150 | 60 |

Effect of Incubation Time on The In Vitro Germination Rate

During the experiment, the pollen germination rate of *G. sinensis* increased as culture time increased. A study was conducted to determine the germination rate and pollen tube length after different incubation times (2 h, 4 h, 8 h, 12 h, 24 h) at 25 °C constant temperature based on the selected liquid culture medium. Its function was to select the culture time suitable for in vitro germination of *G. sinensis* and exclude the difference of pollen germination rate caused by insufficient culture time.

2.4.2. Pollen Viability by Dyeing Methods

The pollen vitality was determined using four staining methods: TTC (2.3.5-Triphenyl Tetrazolium Chloride) staining method [36], MTT (Thiazolyl Blue Tetrazolium Bromide) staining method [37], KI-I$_2$ staining method [38] and Alexander staining method [36]. In vitro germination rate of pollen was used as a control to select the suitable staining methods for *G. sinensis*. The specific operation was as follows: Four kinds of dye were dropped on the slide in advance, and a small amount of pollen was taken with tweezers and mixed with the dye. Following the covering of the glass slide, the following treatments were performed according to different staining methods (Table 3). An optical microscope (Leica DM750) was used to observe and photograph the corresponding slides. A random selection of 5 visual fields with a minimum of 50 pollen grains per visual field was used on each slide, and each method was repeated 6 times. The pollen vitality determined by four staining methods was then calculated based on observation results and the following formula:

**Table 3.** Treatment methods after dyeing.

| Type of Dye Method | Post-Treatment | Judgment Basis of Active Pollen |
| --- | --- | --- |
| TTC | 2 h of dark culture at 37 °C | Red |
| MTT | Stand for 5–10 min | Purple |
| KI-I$_2$ | Observe immediately | Blue |
| Alexander | Observe immediately | cell wall green, protoplast red |

Pollen vitality (%) = number of dyed pollen grains/total number of observed pollen grains × 100.

*2.5. Study on The Pollen Preservation Method of G. sinensis*

2.5.1. Material Selection for Preservation

A large number of fresh male flowers at different flowering stages were collected and brought back to the laboratory in 2022. Then pollen grains isolated from different flowering stages were measured by in vitro germination method to screen suitable pollen preservation materials.

2.5.2. Pollen Storage Conditions

For pollen collection, fresh male flowers were collected based on the selected preservation materials above. The collected pollen was divided into small portions and wrapped into pollen bags by the weighing paper, which was put in a sealed silica gel dryer, drying for 3 h, 6 h and 9 h to reduce the pollen water content to different gradients. After drying, pollen bags of different water contents were placed into 2 mL centrifuge tubes and stored at four different temperatures (RT, 4 °C, −20 °C and −80 °C). After storage, the pollen viability was determined by in vitro germination at 0 day, 7 days, 15 days and 30 days.

*2.6. Statistical Analysis of Data*

The obtained test data were cleared up using Excel 2010. Data were analyzed using one-way analysis of variance, three-factor analysis of variance and orthogonal experimental design in IBM SPSS (version 18.0, IBM, Armonk, NY, USA). R (version 4.1.2) and GraphPad Prism (version8.0.2, Graphpad, San Diego, CA, USA) were used for data visualization.

## 3. Results

### 3.1. Flowering Dynamics Observation of Male Flower

Field studies indicated that the male flowers of *G. sinensis* were in racemes with about 50 small flowers per inflorescence, opening upward from the base (Figure 1a). From the opening to the shed, a single male flower of *G. sinensis* had a short flowering period, lasting 2–3 days. A five-stage dynamic process of the opening of single male flower was artificially categorized: big bud stage (stage I), about 2 h before flower opening, at which the corollas were slightly dilated and the anthers did not extend the corolla, indicating the flowers were about to open (Figure 1b). At the early dispersed-powder stage (stage II), anthers just started dehiscing and dispersing powder. At this time, flowers were half open, with several anthers gathering in the crown and slightly extending out of the corolla. In addition, pollen was visible along the longitudinally split abdominal sutures of the anther (Figure 1c). About 2 h after scattering was the mass scattering stage (stage III), at which the corolla diameter reached its maximum and all anthers cracked loose powder with a large amount of yellow pollens visible to the naked eye. In addition, anthers were yellowish brown and almost perpendicular to the filaments (Figure 1d). At 4 h after dispersed powder was the end dispersed-powder stage (stage IV), during which the corolla began to lose water and the anther color changed to reddish-brown or even black from yellow with some pollen grains still in its dehiscent chamber (Figure 1e). The corolla wipeout stage (stage V) was about 24 h after powder dispersion, where the filaments wilt due to water loss and the corolla was about to fall off (Figure 1f). In brief, male flowers usually have a short single flower life (2–3 days), and the dispersing powder peaks within 4 h following anthers.

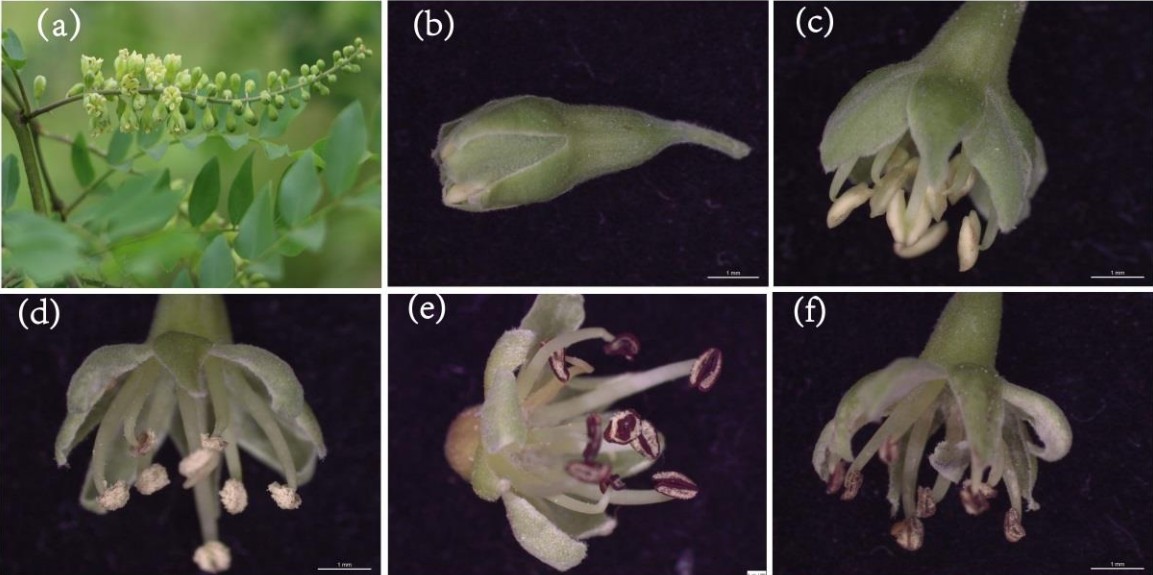

**Figure 1.** Male flowers at different stages: (**a**) Male flower inflorescence, (**b**) Big bud stage, (**c**) Early dispersed pollen stage, (**d**) Mass scattering stage, (**e**) End dispersed pollen stage, (**f**) Corolla wipe-out stage.

### 3.2. Pollen Grain Morphological Characteristics

The results showed that the average pole axis of pollen was 24.03 ± 2.67 um and the average equatorial diameter was 25.29 ± 4.21 um. According to the ratio of its polar axis length (P) to equatorial (P/E = 0.95), pollen grains are oblate spheroidal, belonging to small or medium-sized grains (Figure 2a). Pollen is tricolporate with equatorial elongated apertures (Figure 2b–d), and pollen grains were evenly distributed with thick reticular outer wall ornamentation except for the position close to the margo (Figure 2c,d). The meshes were mostly round, nearly round or irregular polygon among reticular outer wall ornamentation, in which size was irregularly distributed.

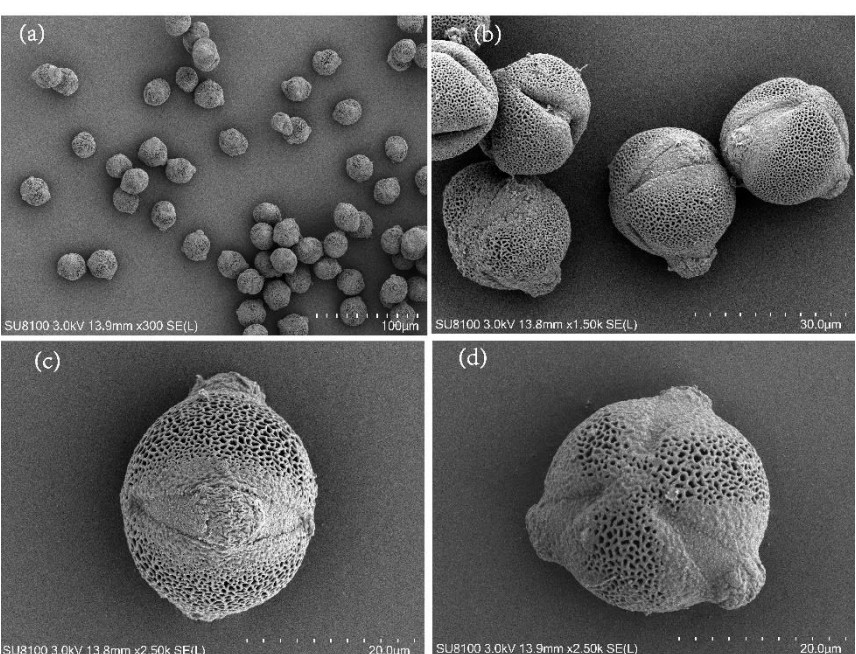

**Figure 2.** Scanning electron microscope (SEM) micrograph of *G. sinensis* pollen grains. Overall shape of pollen grains (**a**,**b**), pollen grain in equatorial view (**c**) and in polar view (**d**).

### 3.3. Research on the Method of Pollen Vitality Determination

3.3.1. In Vitro Germination Method to Determine the Pollen Vitality

Effect of Different Medium Components on In Vitro Germination

The addition of sucrose, $CaCl_2$ and $H_3BO_3$ to the liquid medium significantly affected in vitro germination (Figure 3). The pollen germination increased significantly as the sucrose concentration increased, reaching its highest rate at 15%, but it decreased significantly when sucrose concentrations reached 20%, indicating the appropriate concentration of sucrose promoting pollen germination. The pollen germination rate both decreased with the increase concentration of $CaCl_2$ and $H_3BO_3$ if the concentration was more than 100 mg/L, which showed that more calcium chloride and boric acid were not conducive to the pollen germination.

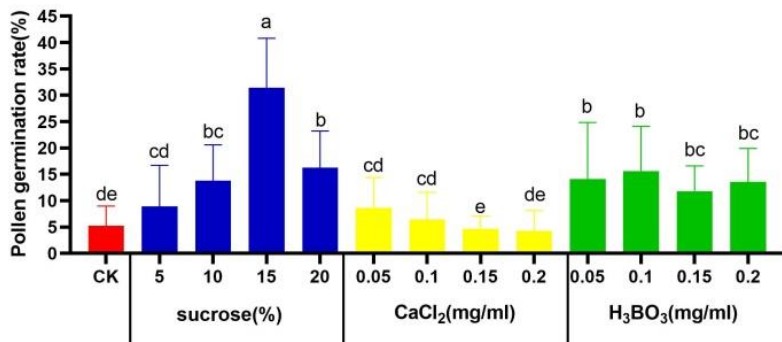

**Figure 3.** Effect of different concentration of Sucrose, $CaCl_2$ and $H_3BO_3$ on in vitro pollen germination of *G. sinensis*.

To sum up, the optimum concentrations of the sucrose, $CaCl_2$ and $H_3BO_3$ were 15% (30.90%), 50 mg/L (9.22%) and 100 mg/L (17.15%), respectively, all significantly higher than CK (5.21%). Too low or too high concentration would reduce the pollen germination rate of *G. sinensis*.

Selection of Liquid Culture Medium Formula

The results of an orthogonal test design and its data variance analysis (Table 4) indicated that sucrose (F = 106.147, $p < 0.001$) was most effective in the germination of pollen, followed by calcium chloride (F = 15.786, $p < 0.001$) and boric acid (F = 42.304, $p < 0.001$). During this experiment, 15% sucrose, 20 mg/L calcium chloride and 100 mg/L boric acid were found to be the best combination of medium. The pollen germination rate increased significantly with the increase of sucrose concentration (Figure 4). In response to boric acid concentration, pollen germination rate varied. B2 was significantly higher than B1 and B3, but B1 and B3 did not differ significantly, indicating that 100 mg/L boric acid was more beneficial to the in vitro germination. Pollen germination was also affected by the concentration of $CaCl_2$. A significant difference did not exist between C1 and C3, but both were significantly higher than C2, indicating that 20 mg/L and 40 mg/L $CaCl_2$ significantly promoted pollen germination.

**Table 4.** The range analysis of germination rates using an Orthogonal Assay Test Strategy (OATS). L Combinations (Levels Factors) = L9 (34) on different media.

| Combinations | Sucrose (%) A | Boric Acid (mg/L) B | Calcium Chloride (mg/L) C | Empty Column | Pollen Germination Rate |
|---|---|---|---|---|---|
| 1 | S1 | B1 | C1 | 1 | 25.50% ± 0.55% |
| 2 | S1 | B2 | C2 | 2 | 19.65% ± 0.46% |
| 3 | S1 | B3 | C3 | 3 | 19.97% ± 0.22% |
| 4 | S2 | B1 | C2 | 3 | 17.68% ± 0.27% |
| 5 | S2 | B2 | C3 | 1 | 33.36% ± 0.31% |
| 6 | S2 | B3 | C1 | 2 | 23.85% ± 0.36% |
| 7 | S3 | B1 | C3 | 2 | 41.20% ± 0.21% |
| 8 | S3 | B2 | C1 | 3 | 47.40% ± 0.5% |
| 9 | S3 | B3 | C2 | 1 | 29.71% ± 0.27% |
| K1 | 65.12% | 84.38% | **96.75%** | 88.57% | |
| K2 | 74.89% | **100.41%** | 67.04% | 84.70% | |
| K3 | **118.31%** | 73.53% | 94.53% | 85.05% | |
| x̄1 | 21.71% | 28.13% | **32.25%** | 29.52% | |
| x̄2 | 24.96% | **33.47%** | 22.35% | 28.23% | |
| x̄3 | **39.44%** | 24.51% | 31.51% | 28.35% | |
| R | 53.19% | 26.88% | 29.71% | 3.87% | |
| Order of influencing factors | | | A > C > B | | |
| Best levels | A3 | B2 | C1 | | |
| Excellent combination | | | A3 B2 C1 | | |

Ki is the sum of the different levels for the i th factor, xi is the mean of the different levels for the ith factor, and R the range difference between the max and mini values. Three levels of sucrose (S): S1 = 5 g/100 mL, S2 = 10 g/100 mL, S3 = 15 g/100 mL; boric acid (B): B1 = 50 mg/L, B2 = 100 mg/L, B3 = 150 mg/L; Calcium nitrate (C): C1 = 20 mg/L, C2 = 40 mg/L, C3 = 60 mg/L. Also see Table 3. Bold numbers are the highest values and bold letters are the best components.

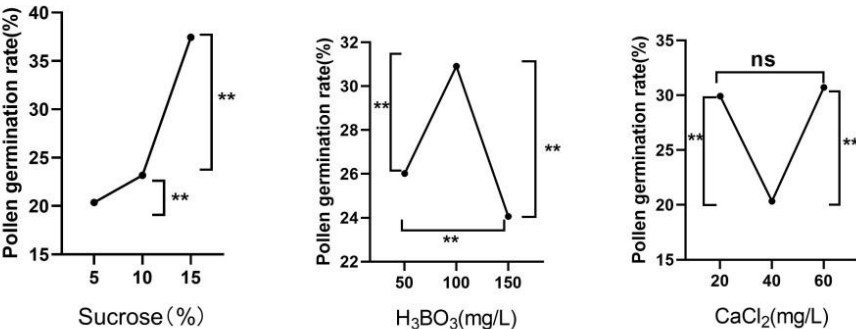

**Figure 4.** Effect of combination of sucrose, $H_3BO_3$ and $CaCl_2$ on pollen germination of *G. sinesis* in liquid medium, ns—non-significant; ** significant at $p \leq 0.01$.

In Vitro Germination Rate of Different Culture Durations

A significant increase in pollen germination rate and pollen tube length was observed with prolonged incubation (Figure 5A) when pollen grains were cultured in the liquid medium (15% sucrose concentration + 20 mg/L calcium chloride + 100 mg/L boric acid) at a constant temperature of 25 °C. As a result, the highest germination rate was 65.89% ± 3.41% (Figure 5B), and the pollen tube reached to 3.96 mm ± 0.06 mm after 24 h in culture. The observation results of pollen tube dynamics indicate that the pollen tube along the style path reached the ovule 24 h after pollination (unpublished). Therefore, in this study, the germination rate of pollen cultured in liquid medium for 24 h was taken as the final pollen germination rate of *G. sinensis*. Thus, the vitality of fresh pollen was 65.89% ± 3.41%.

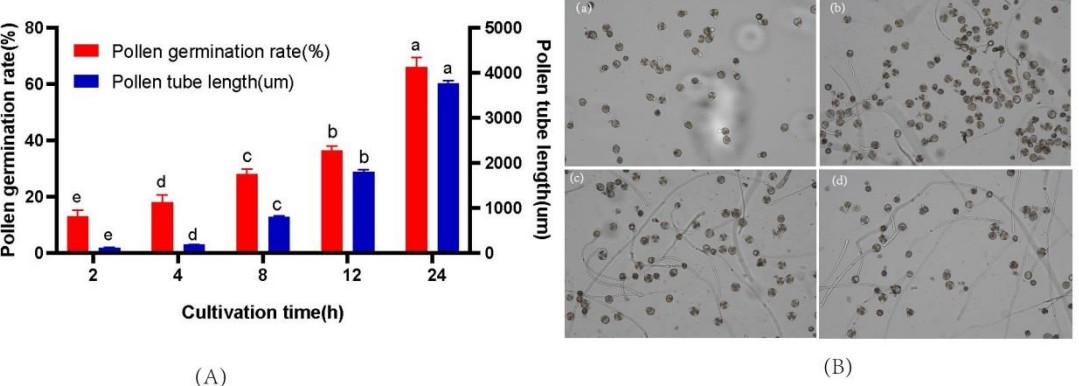

(A)

(B)

**Figure 5.** (**A**) Effect of incubation time on in vitro pollen germination and tube growth; (**B**) The four different photos show the pollen germination and pollen tube growth status at different time of incubation (a) 1 h, (b) 4 h, (c) 12 h, and (d) 24 h.

3.3.2. Pollen Vitality of *G. sinensis* Measured by Four Different Staining Methods

In three of four dyeing methods, active pollen could be differentiated from inactive pollen except in KI-$I_2$, which was the only dyeing method that could not distinguish between active and inactive pollens (Figure 6d). The MTT staining method, TTC staining method and Alexander staining method revealed pollen vitality of 68.19%, 36.87% and 90.40%, respectively, and they differed significantly from one another (Table 5). Comparing the in vitro germination rate with dyeing methods allowed us to decide that the MTT staining method was suitable for the rapid determination of the pollen vitality of *G. sinensis*. As compared to a control sample, MTT staining method did not significantly alter pollen vitality.

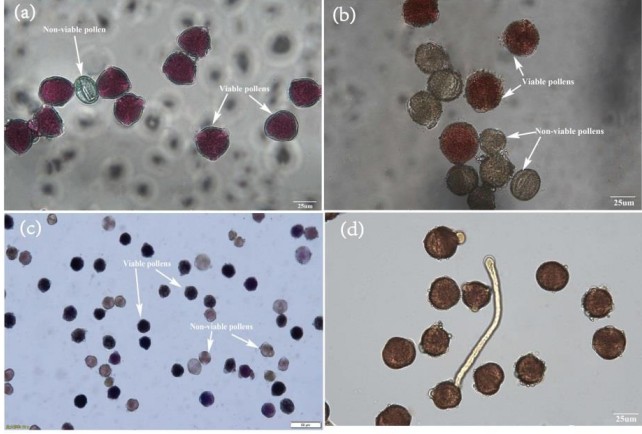

**Figure 6.** Pollen grains were stained by four staining method, (**a**) Alexander staining method, (**b**) TTC staining method, (**c**) MTT staining method and (**d**) KI-$I_2$ staining method.

**Table 5.** Comparison of pollen viability results determined with four methods.

| Methods | | Pollen Viability ± Standard Error |
|---|---|---|
| In vitro germination staining | | 65.89% ± 3.41% b |
| | TTC | 36.87% ± 1.80% c |
| | Alexander | 90.40% ± 1.54% a |
| | MTT | 68.19% ± 2.09% b |

A significant difference between methods was indicated by lowercase letters ($p < 0.05$). Values are means ± SE.

### 3.4. Pollen Preservation Method for G. sinensis

3.4.1. Effect of Pollens at Different Flowering Stages on In Vitro Germination Rate

The results showed that pollen germination rates increased gradually with incubation time for four flowering stages (Figure 7). In addition, the final pollen germination rate among four stages was not significantly different after 24 h of pollen culture. Based on these results, the pollen viability and germination trend were basically the same for the four stages. Despite this, the germination rate of pollen at different flowering stages varied significantly from 2–4 h, with higher germination rate of the fourth stage and lower germination rate of stage 2. This might be explained by the different water content of pollen grains during dispersion due to their brief dormancy. As a result, pollens of stage 2 were best suited to instant pollination since all anthers had cracked and there were a large number of pollen grains exposed. On the basis of the above results, pollen at the large bud stage can be collected to improve pollen collection efficiency without worrying about pollen viability.

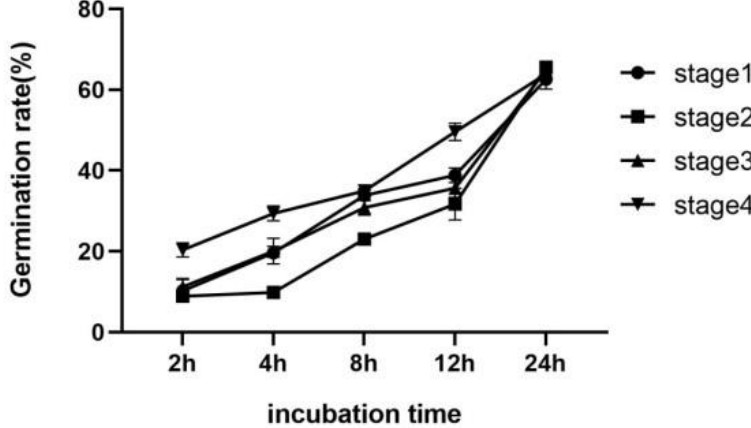

**Figure 7.** Effects of pollen at different flowering stages on pollen germination rate.

3.4.2. Short-Term Preservation of Pollen

Using the three-factor ANOVA method, the results indicated that storage time (F = 2298.6, $p < 0.0001$), storage temperature (F = 891.01, $p < 0.0001$) and pollen water content (F = 17.45, $p < 0.0001$) were significant factors affecting pollen germination. In addition, there are two interactions among the three influencing factors: storage time * storage temperature (F = 164.49, $p < 0.0001$), storage time * pollen water content (F = 38.19, $p < 0.0001$), storage temperature * pollen water content (F = 13.81, $p < 0.0001$). The interaction of three factors, storage time * storage temperature * pollen water content (F = 11.86, $p < 0.0001$), also existed.

Effect of Pollen Water Content on In Vitro Germination

After 3 h, 6 h and 9 h of silica gel drying, pollen water content was reduced to 14%, 9% and 5%, and the germination rate of pollens were 59.03% ± 3.20%, 57.97% ± 2.02% and 52.13% ± 3.47%, which were lower than that of fresh pollens by 10.41%, 12.02% and 20.88%, respectively. Based on the results above, pollen germination was greatly affected by the pollen water content and decreased significantly with decreasing water content.

The pollen preservation results showed all pollen germination rates under RT treatment were almost zero after 30 days of storage (Figure 8). When stored at 4 °C for 30 days, pollen with a higher water content of 14% had a significantly lower germination rate than that with a lower water content of 9% and 5%. It meant that pollen could be preserved for longer if its water content was reduced when stored under 4 °C. With a reduction in storage temperature to −20 °C or −80 °C, there was no significant difference in pollen germination rates with different water contents after 30 days, but after 30 days, the germination rate of pollen stored at −20 °C was much lower than that of those stored at −80 °C. In spite of the significant effect of pollen water content on the germination rate after storage, the effect gradually weakened with the storage temperature decreased. This indicated that the water content of pollen (5%–14%) was no longer the main factor affecting the germination rate after storage, if the storage temperature reduced to −20 °C or below.

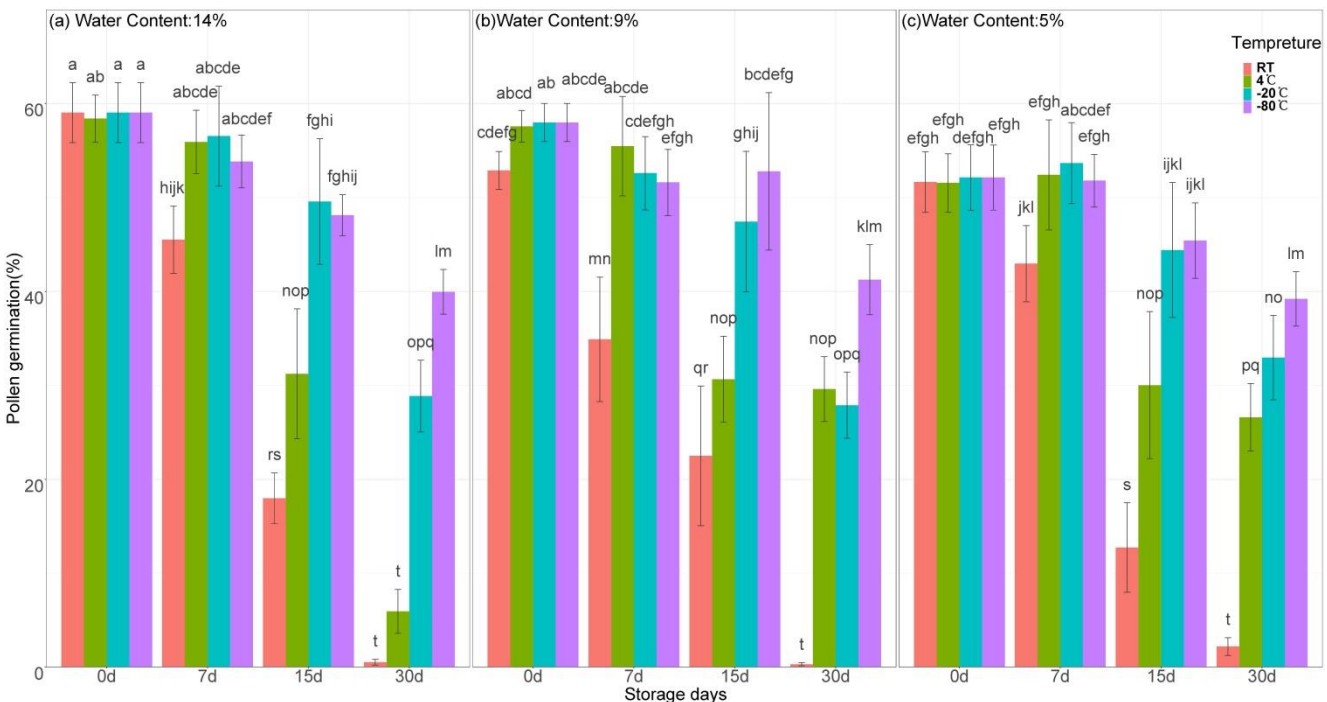

**Figure 8.** Effects of storage temperature, storage time and pollen water content on the pollen preservation of *G. sinensis*. A significant difference between methods was indicated by lowercase letters (*p* < 0.05) using Multiple Testing.

Effect of Storage Temperature on In Vitro Germination

As the storage time increased, the germination rate of all treatments decreased significantly. The pollen germination rate was almost zero after 30 days of storage under RT treatment. However, pollen stored at 4 °C, −20 °C and −80 °C were treatments that were effective in delaying the decreasing trend of pollen germination, and a temperature of −80 °C was most effective for preserving pollen.

Effect of Storage Time on In Vitro Germination

Despite the aforementioned observations, all treatments showed a gradual decline in pollen germination with the extension of storage time (Figure 8). Pollen germination rates of different treatments decreased at different rates due to the effects of storage temperature, pollen water content, storage time and interactions between them. Pollen germination decreased the fastest under RT treatment. Within 7 days of storing, the germination rate of pollen stored at other low temperatures remained similar to that before storage, except for RT treatment. This indicated that the pollen germination rate remained unchanged at low temperature after short-term storage for 7 days. Although the pollen germination rate

decreased with the extension of the storage time to 15 or 30 days, the lower the storage temperature, the higher the pollen germination rate. That is to say that storage temperature gradually became the main factor affecting the germination rate of pollen if the storage time was over 7 days.

The highest germination rate was observed after 30 days of cryopreservation at $-80\,°C$ for *G. sinensis* pollen, if the pollen water content was reduced to 9%, which was only 28.84% lower than that before storage. This could be used as a suitable method for preserving pollen.

## 4. Discussion

### 4.1. Pollen Morphological Characteristics of G. sinensis

Palynology is often used in plant classification and phylogenetic studies [39–41], the size of pollen grains according to the appearance and shape of pollen [41], pollen outer wall ornamentation [40,42] and number and location of germination pores [43]. Observations of the microscopic morphology of pollen revealed that pollen grains of *G. sinensis* were oblate spheroidal, tricolporate with equatorial elongated endoapertures and a reticulate sporoderm surface. This is consistent with its classified status because pollen of the majority of *caesalpinioid* species is isopolar and tricolporate with perforate or reticulate surface ornamentation, according to Hannah Banks & Paula J. Rudall [44]. The small or medium pollen size is also consistent with the widespread type of pollen in *caesalpinioid*, in which the widespread type of pollen is small to medium in size [45]. The equatorial elongated endoapertures may be a strategy help to preserve the structural integrity of the pollen wall and, therefore, the viability of the pollen grain [44]. Endoapertures have been found to be positively correlated with pollen germination rate in studies [46]. The pollen of *G. sinensis* germinated rapidly after 1 h in culture, consistent with this. In this article, pollen grains have reticulate ornamentation (with sculpturing elements forming an open network or reticulum over the pollen surface), which was related to foldable structures and the natural design of pollen grains during pollen dehydration [47]. In short, the shape and appearance of pollen and pollen apertures are closely related to the classification of this species [44,45].

### 4.2. Methods for the Determination of Pollen Viability

In vitro germination was considered to accurately predict pollen viability [11,48–50]. Due to the specificity of plant species, the most important step of in vitro germination was selecting the appropriate medium for pollen germination. An addition of 15% sucrose, 100 mg/L boric acid and 20 mg/L calcium chloride enhanced pollen germination in this study. This was consistent with the existing research results because sucrose was the most effective carbon source [51,52] and an important signaling substance during the in vitro pollen germination [53]. As well as promoting pollen germination, boron contributes to pollen tube elongation through signal transduction [54,55]. The mineral calcium ($Ca^{2+}$) regulates ion balance and plays a critical role in pollen tube growth [14,56].

The concentration of sucrose required for pollen germination varied with plant species [50,57]. Similarly to previous studies in other plants [58,59], 15% sucrose added to pollen germination of *G. sinensis* was most beneficial. According to the single factor experiment, variations in boron concentrations did little to promote pollen germination since the amount of boron required in the pollen medium had the smallest effect [60]. A slight deviation could cause insufficient [54] or excessive toxicity [61], which was consistent with the results of previous Fragallah [62,63]. Moreover, 50 mg/L $CaCl_2$ increased pollen germination, but the pollen germination rate decreased with the increase of $CaCl_2$ concentration. This may be related to the regulation of pollen tube calcium signal [13] and the inhibition of pollen tube growth by high $Ca^{2+}$ concentration [64].

In order to detect pollen viability, the staining method was considered to be the easiest and fastest method [38,65,66]. The pollen viability of *G. sinensis* was assessed using four staining methods, and it varied with different methods. In comparison to in vitro germi-

nation rate, TTC's pollen viability was lower while Alexander's was higher. KI-$I_2$ did not distinguish non-viable pollen. Only MTT was a suitable method for accurate determination of pollen viability of *G. sinensis.* A major reason for the difference between four staining methods was the specificity of pollen [36,67]. The results of this study were similar to those of other pollen viability studies [10,38,57]. TTC reacts with dehydrogenase in normal pollen and appears red, which is considered to be the most common technique [68]. However, it is a fat-soluble photosensitive complex, susceptible to environmental influences. In this article, it did not stain pollen grains of *G. sinensis* well, just as the results described in *Sinobeam* [69] and *Rhododendron* [37], which may be caused by the failure to meet the conditions due to operating errors. In spite of the fact that Alexander's staining method was able to distinguish viable pollen grains and its result was much higher than those of the control, the reason of it may be attributed to weak ability to distinguish pollen viability. The KI-$I_2$ staining method relies on starch to stain pollen. Moreover, the staining effect was affected not only by the amount of starch, but also by the ratio of amylopectin to amylose. However, pollen with vitality cannot be determined using this method in this study due to all pollen grains stained with the same color. Viable pollen grains could be stained to purple by MTT staining method based on the activity of dehydrogenase in pollen grains. This method is suitable for the determination of pollen viability of *G. sinensis* in this study, just as *Rhododendron* [37].

*4.3. Pollen Preservation of G. sinensis*

In certain storage conditions, such as low temperature, low oxygen and low humidity, enzyme activity and related metabolic activities may be reduced or inhibited, thereby extending pollen grain life [70–74]. It was found that the pollen germination rate was easily affected by storage temperature, storage time and pollen water content during pollen storage of *G. sinensis* in this study. A common feature of all pollen preservation treatments was a decrease in germination rate with increasing storage time, such as plum [75], pecan [49], rose [15], litchi [76] and chrysanthemum [77]. It was almost impossible for pollen stored at RT to germinate after 30 days, but it was still possible to maintain pollen vitality at low temperatures. Especially when preserved at −80 °C, the germination rate only decreased by 28.84%, which was the same as the results of previous studies on hickory walnut and chrysanthemum [57,76]. The three water contents of pollens set in this study played a certain role in the pollen preservation, but when the temperature was reduced to −20 °C or below, pollen viability was no longer largely determined by the pollen water content. It may be caused by an insufficient gap between water content gradients [71,75]. A suitable method for short-term pollen preservation of *G. sinensis* was identified in this study. However, further pollination tests were needed to more accurately evaluate the feasibility of this preservation method in the breeding of *G. sinensis*.

**5. Conclusions**

The study results showed that male flowers of *G. sinensis* lasted 2–3 days during their single flowering period. The dynamic process of single male flower opening was artificially divided into five stages. Pollen grains of *G. sinensis* are oblate spheroidal, tricolporate with equatorial elongated endoapertures and the sporoderm surface is reticulate. The fresh pollen germination rate of *G. sinensis* reached its maximum value (65.89% ± 3.41%), and the length of pollen tube was 3.96 mm after 24 h incubation at 25 °C based on the selected liquid medium: 15% sucro e + 100 mg/L boric acid + 20 mg/L calcium chloride. In comparison with the other three staining methods, the MTT staining method could accurately and rapidly determine pollen viability. A short-term preservation method for *G. sinensis* can be achieved by reducing the water content of pollen to 9% and then storing it at −80 °C for 30 days.

**Author Contributions:** Q.L. and J.Y. designed and carried out the experiments. Q.L. performed the data analyses and drafted the manuscript. X.W. and Y.Z. supported this research and made revisions to the manuscript. All authors have read and agreed to the published version of the manuscript.

**Funding:** This work was supported by the Characteristic Forestry Industry Research Project of Guizhou Province (GZMC-ZD20202098); Science and Technology Plan Project of Guizhou Province ([2020]1Y056); Guizhou Characteristic Forestry Industry scientific Research Project-Research and Demonstration of Key Technology of Directional Cultivation of Gleditsia sinensis, No. Te LinYan: 2020.9-2023.12; Guizhou Science and Technology Planning Project (Guizhou Science and Technology Cooperation Support [2020]1Y058).

**Data Availability Statement:** The data presented in this study are available on request from the corresponding author.

**Acknowledgments:** The author would like to thank College of forestry and the Institute of Forest Resources & Environment of Guizhou to provide Laboratory platform and experiment management teachers for their assistance with microscopy.

**Conflicts of Interest:** The authors declare no conflict of interest.

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
