# Peer review of "Studies on Pollen Morphology, Pollen Vitality and Preservation Methods of Gleditsia sinensis Lam. (Fabaceae)"

_forests, doi:10.3390/f14020243_

Round 1

Reviewer 1 Report

The manuscript gives a veery interesting insight to the morphology, pollen viability and preservation methods of pollen from Gleditsia sirensis.

The plant is also seen in Europe as an ornamental shrub in gardens.

The authors are asked to add a sentence in the introduction part about the possible allergenicity of pollen for humans - if data on this are available.

Author Response

I am so sorry for not satisfyng your request , because I do not have data on the sensitization of this pollen to humans.

Reviewer 2 Report

See comments in the manuscript 

Reviewer 3 Report

Dear authors,

The manuscript "Studies on Pollen Morphology, Pollen Vitality and Preservation Methods of Gleditsia sinensis" needs further revisions*, mainly in relation to the description of the pollen morphology of the species.

*All considerations, suggestions and/or corrections are highlighted in the attached PDF file.

1 – Abstract: This description of pollen morphology is quite informal. Use terminology appropriate to the field of study. Look at comments during the manuscript.

2 - Introduction: The topic “Introduction” needs to be more complete in information about the taxonomy of the species. What hierarchical levels does the species belong? Which family? There is no information about this. Why this species? What is the relevance of studying this species in the group as a whole? I suggest adding a paragraph on these issues, especially in relation to taxonomy. This is information that cannot be missing from this topic.

3 - Materials and Methods: What measurements were performed in this study? How many pollen grains were measured? What references were used to describe pollen grains? These are important informations that must be present in the M&M.

4 - Results: The description of pollen morphology is quite inappropriate. The title of the article says “...pollen morphology...” and therefore, it is expected that there will be a description with adequate terminology of the study field. I suggest reading classic glossaries of Palynology, I recommend: Erdtman (1952), Punt et al. (2007) and Halbritter et al. (2018), as well as, for order of description of pollen morphology, the article by Bellonzi et al. (2020).

5 - Discussion: Improve the discussion according to the suggestions that were mentioned above in relation to the description of the pollen morphology.

Round 2

Reviewer 2 Report

see comments in the manuscript

Reviewer 3 Report

Dear authors,

The manuscript "Studies on Pollen Morphology, Pollen Vitality and Preservation Methods of Gleditsia sinensis" has been revised again and at this moment, there are only a few comments about questions, definitions and references corrections, and these are highlighted in the attached PDF file.

*All considerations, suggestions and/or corrections are highlighted in the attached PDF file.
